# Intramolecular feedback regulation of the LRRK2 Roc G domain by a LRRK2 kinase-dependent mechanism

Bernd K Gilsbach[1], Franz Y Ho[2], Benjamin Riebenbauer[1], Xiaojuan Zhang[2], Giambattista Guaitoli[1], Arjan Kortholt[2,3]*, Christian Johannes Gloeckner[1,4]*

[1]German Center for Neurodegenerative diseases (DZNE), Tübingen, Germany; [2]Department of Cell Biochemistry, University of Groningen, Groningen, Netherlands; [3]YETEM-Innovative Technologies Application and Research Centre, Suleyman Demirel University West Campus, Isparta, Turkey; [4]Core Facility for Medical Bioanalytics, Institute for Ophthalmic Research, Center for Ophthalmology, University of Tübingen, Tübingen, Germany

## eLife assessment

This **valuable** article reports on the relationship between GTP hydrolysis parameters and kinase activity of LRRK2, which is associated with Parkinson's disease. The authors provide a detailed accounting of the catalytic efficiency of the ROC GTPase domain of pathogenic variants of LRRK2, in comparison with the wild-type enzyme. The authors propose that phosphorylation of T1343 inhibits kinase activity and influences monomer–dimer transitions, but the experimental evidence is currently **incomplete**.

**\*For correspondence:**
a.kortholt@rug.nl (AK);
johannes.gloeckner@dzne.de
(CJG)

**Competing interest:** The authors declare that no competing interests exist.

**Abstract** The Parkinson's disease (PD)-linked protein Leucine-Rich Repeat Kinase 2 (LRRK2) consists of seven domains, including a kinase and a Roc G domain. Despite the availability of several high-resolution structures, the dynamic regulation of its unique intramolecular domain stack is nevertheless still not well understood. By in-depth biochemical analysis, assessing the Michaelis–Menten kinetics of the Roc G domain, we have confirmed that LRRK2 has, similar to other Roco protein family members, a $K_M$ value of LRRK2 that lies within the range of the physiological GTP concentrations within the cell. Furthermore, the R1441G PD variant located within a mutational hotspot in the Roc domain showed an increased catalytic efficiency. In contrast, the most common PD variant G2019S, located in the kinase domain, showed an increased $K_M$ and reduced catalytic efficiency, suggesting a negative feedback mechanism from the kinase domain to the G domain. Autophosphorylation of the G1+2 residue (T1343) in the Roc P-loop motif is critical for this phosphoregulation of both the $K_M$ and the $k_{cat}$ values of the Roc-catalyzed GTP hydrolysis, most likely by changing the monomer–dimer equilibrium. The LRRK2 T1343A variant has a similar increased kinase activity in cells compared to G2019S and the double mutant T1343A/G2019S has no further increased activity, suggesting that T1343 is crucial for the negative feedback in the LRRK2 signaling cascade. Together, our data reveal a novel intramolecular feedback regulation of the LRRK2 Roc G domain by a LRRK2 kinase-dependent mechanism. Interestingly, PD mutants differently change the kinetics of the GTPase cycle, which might in part explain the difference in penetrance of these mutations in PD patients.

## Introduction

Nonsynonymous sequence variants within the Leucine-Rich Repeat Kinase 2 (LRRK2) are associated with familial forms of Parkinson's disease (PD) and are phenotypically indistinguishable from idiopathic forms of PD (iPD) (*Paisán-Ruíz et al., 2004*; *Zimprich et al., 2004*). In addition to defined pathogenic as well as PD-risk coding variants, the LRRK2 locus also contributes to an increased risk of developing iPD more generally, as shown in genome-wide association studies (*Simón-Sánchez et al., 2009*). LRRK2 is a multidomain protein, which belongs to the Roco protein family (*Marín et al., 2008*). Besides a core module, conserved among the Roco proteins, consisting of a Roc (Ras of complex proteins) G domain and a COR (C-terminal of Roc) dimerization domain, which is followed by a kinase domain, it contains four predicted solenoid domains. The three N-terminal solenoid domains consist of Armadillo repeats followed by an Ankyrin domain and the namesake leucine-rich repeats, while the C-terminus is formed by a seven-bladed WD40 domain (*Mills et al., 2012*). By similarity, these domains are involved in protein–protein interactions and, based on recent structural models and high-resolution structures, are likely to be involved in the intramolecular regulation of the protein, as well as, for example, by keeping the kinase domain in an auto-inhibited state (*Deniston et al., 2020*; *Gloeckner and Porras, 2020*; *Guaitoli et al., 2016*; *Taylor et al., 2020*). Moreover, membrane-bound LRRK2 has been demonstrated to phosphorylate a specific set of Rab proteins at a conserved threonine residue within their switch II motif, including Rab8a and Rab10 (*Gomez et al., 2019*; *Steger et al., 2016*). In addition, LRRK2 kinase activity has been demonstrated to be involved in endo-lysosomal pathways, thereby being a regulator of the innate immunity (*Ahmadi Rastegar and Dzamko, 2020*; *Bonet-Ponce and Cookson, 2019*; *Erb and Moore, 2020*). Despite these findings, the molecular mechanisms underlying LRRK2 activation and regulation are still not completely understood.

It is clear that the kinase activity of LRRK2 is dependent on the guanine nucleotide-binding capacity of the Roc domain. Despite the availability of high-resolution structures, the complete mechanism of G-nucleotide action remains to be determined, in particular if the nucleotide state of the Roc domain influences kinase activity (*Biosa et al., 2013*; *Ito et al., 2007*; *Taymans et al., 2011*).

Previous work on the conserved RocCOR module of a bacterial Roco protein has suggested that the nucleotide state of the Roc domain regulates dimerization (*Deyaert et al., 2017*). Consistently, also accumulating data shows that a monomer dimer/oligomer transition plays a critical role in LRRK2 protein activation (*Berger et al., 2010*; *Schapansky et al., 2014*; *Wu et al., 2006*). Cellular data suggest that LRRK2 shuttles between a monomeric cytosolic states and kinase-active oligomeric forms localized at the membrane (*Berger et al., 2010*; *James et al., 2012*). Further support comes from recent high-resolution structures, indicating that a Rab29-induced tetramerization represents the kinase active state (*Wu et al., 2006*). Our recent data revealed that autophosphorylation induces LRRK2 monomerization (*Guaitoli et al., 2023*). Interestingly, the Roc domain has been identified as a major target of LRRK2 autophosphorylation by phospho-proteomic analysis (*Gloeckner et al., 2010*; *Greggio et al., 2009*; *Webber et al., 2011*). Furthermore, there are indications that autophosphorylation of the Roc G domain modifies GTP-binding activities (*Webber et al., 2011*) and that trans-phosphorylation by the kinase domain of the LRRK2 orthologue *Dictyostelium* Roco4 enhances GTPase activity (*Liu et al., 2016*). In this study, we analyzed the GTPase activity of LRRK2 and its PD mutants in depth and by combining systematic mutational analysis with the determination of the Michaelis–Menten kinetics. Based on this analysis, we identified an intramolecular negative feedback loop defined by LRRK2 autophosphorylation and its impact on the monomer–dimer equilibrium and GTPase activity.

## Results

### Impact of PD variants on LRRK2 GTPase activity

We and others have previously determined a full Michaelis–Menten kinetics for the GTPase hydrolysis mediated by Roco proteins as well as for wild-type LRRK2 (*Liu et al., 2010*; *Wauters et al., 2018*). LRRK2 has kinetic parameters comparable to other Roco protein family members. Strikingly, the $K_M$ value of LRRK2 lies within the range of the physiological GTP concentrations (around 500 µM; *Traut, 1994*) within the cell. Therefore, changes in global cellular or local GTP concentration might have a high impact on protein activity. For this reason, we here fully assessed the enzyme kinetics of disease-associated PD variants, in particular those with confirmed disease co-segregation. For this purpose,

we selected three pathogenic variants, R1441G, localized within the Roc domain, the Y1699C variant in the COR-A domain, as well as the kinase-domain variant G2019S. All three variants significantly increase the phosphorylation of the LRRK2 substrates Rab8a, Rab10, and Rab12 (*Kalogeropulou et al., 2022*). In addition to full-length LRRK2 expressed in HEK293T cells, we included a LRRK2 RocCOR domain construct expressed as an MBP fusion protein from *Escherichia coli* in this study. Besides the wild-type, the variants R1441G and Y1699C have been successfully expressed and purified from *E. coli*. For full-length LRRK2, only the Roc-domain variant R1441G as well as the kinase-domain variant G2019S could be analyzed due to considerable low expression and stability of a full-length LRRK2 Y1699C construct in HEK293T cells. LRRK2 wt, R1441G, as well as Y1699C MBP-RocCOR constructs showed similar kinetic parameters in the standard radiolabeling assay (charcoal assay), which determines the release of free inorganic phosphate (Pi) (*Kolch, 2000*). While slight differences were observed in the individual $K_M$ as well as $k_{cat}$ values, the catalytic efficiency determined by $k_{cat}/K_M$ was identical (*Figure 1A–D*; *Supplementary file 1a and b*). For full-length LRRK2, we have chosen an HPLC-based assay to overcome limitations in the amount of protein necessary to determine a full Michaelis–Menten kinetics. This assay determines the production of Guanosine Diphosphate (GDP) by chromatographic separation of the nucleotides (*Ahmadian et al., 1997*). To ensure that both assays result in comparable results, we compared GTP hydrolysis rates ($k_{obs}$) of full-length LRRK2 wt in both assays at two different GTP concentrations. Both assays, the HPLC-based assay as well as the charcoal assay, resulted in overall comparable kinetics, ensuring that the results of both assays can be directly compared (*Figure 1E and F*, *Table 1*). For the wild-type full-length LRRK2 protein, we have determined a $k_{cat}$ of 0.36 ± 0.02 min⁻¹ and a $K_M$ of 554 ± 62 µM confirming our previously published data, thus demonstrating the robustness of the used HPLC-based assay (*Wauters et al., 2018*).

Interestingly, full-length LRRK2 bearing the R1441G variant showed a decreased Michaelis–Menten constant ($K_M$ = 271 ± 27 µM) while the $k_{cat}$ increased compared to the wt (*Figure 2A*), leading to an augmented catalytic efficiency of that variant. In contrast, full-length G2019S LRRK2 showed a significantly increased $K_M$ value of 867 ± 110 µM compared to a wt reference (554 ± 62 µM), while no significant change in the $k_{cat}$ parameter ($k_{cat}$ = 0.36 min⁻¹) was observed (*Figure 2B*). Together, our data show that LRRK2 has a $K_M$ value that lies within the range of the physiological GTP concentrations and full-length G2019S LRRK2 showed a significantly increased $K_M$ value.

## Autophosphorylation regulates GTPase activity

Since the pathogenic G2019S variant has increased kinase activity, we next tested if a kinase-inactive variant has the opposite effect on GTPase activity. In fact, the $K_M$ value ($K_M$ = 181 ± 58 µM) for a K1906M kinase-dead LRRK2 variant is approximately three times lower compared to wt LRRK2 while the $k_{cat}$ value was only slightly reduced (*Figure 2C*). Consistently, MLi-2 treated fl. wt LRRK2 also has a lower $K_M$ value (*Figure 2—figure supplement 1*), which also ensures that the GTPase and not the kinase domain is the driver of the GTP consumption in our experimental setup (*Liu and West, 2017*). This strongly suggests a negative feedback signal on the Roc domain conferred by an active kinase conformation, most likely mediated via autophosphorylation. To further test this hypothesis, we incubated wild-type LRRK2 with ATP, in vitro. In fact, an incubation of the full-length wild-type protein with ATP for 2 hr at 30°C prior to determining its GTPase activity leads to altered Michaelis–Menten parameters, impacting the $K_M$ as well as $k_{cat}$ values. An approximately twofold increase in $K_M$ (1036 ± 168 µM) and an approximately twofold decrease in $k_{cat}$ (0.13 ± 0.01 min⁻¹) demonstrate a strong reduction in LRRK2 GTPase activity by enforced autophosphorylation (*Figure 3A*). As a control condition, to rule out inactivation of the LRRK2 protein by degradation during pre-incubation, we incubated a dead variant (K1906M) with ATP, the same way as the wild-type, resulting in no significant change in $K_M$ and $k_{cat}$ values. This shows that autophosphorylation indeed negatively regulates the GTPase activity of LRRK2.

## The P-loop site T1343 is a critical site for LRRK2 phosphoregulation

To identify the phosphosites that are important for the feedback mechanism, we tested several serine/ threonine to alanine mutants (*Figure 3B*), expecting that the critical autophosphorylation site would show no change in $k_{cat}$ and $K_M$ values for GTP hydrolysis upon in vitro ATP treatment. By mass-spectrometric mapping, we and others have previously demonstrated that the Roc domain is a hub for LRRK2 autophosphorylation (*Gloeckner et al., 2010*; *Greggio et al., 2009*; *Webber et al., 2011*).

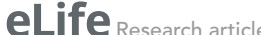

**Figure 1.** Determination of kinetic parameters for LRRK2 GTP hydrolysis of Parkinson's disease (PD) variants by the charcoal assay. (**A**) Michaelis–Menten kinetics for pathogenic variant within the RocCOR module. (**B**) Comparison of $K_M$ values (n: wt=5, R1441G=6, Y1699C=5). (**C**) Comparison of $k_{cat}$ values (n: wt=5, R1441G=6, Y1699C=5). (**D**) Catalytic efficiency ($k_{cat}/K_M$) (n: wt=5, R1441G=6, Y1699C=5). (**E–G**) Determination of $k_{obs}$ values for full-length LRRK2 at 100 µM (**E**) and 2000 µM (**F**) GTP (both: n=4). (**H**) Domain structure of LRRK2 and the position of PD variants analyzed in this study.

**Table 1.** HPLC-based full-length LRRK2 Michaelis–Menten kinetics.

| LRRK2 variant | $K_M$ (μM) | $k_{cat}$ (min⁻¹) | $k_{cat}/K_M$ (min⁻¹/mM) |
|---|---|---|---|
| wt | 554 ± 62 | 0.36 ± 0.02 | 0.71 ± 0.08 |
| wt+ATP | 1036 ± 169 | 0.14 ± 0.01 | 0.13 ± 0.02 |
| R1441G | 272 ± 28 | 0.43 ± 0.02 | 1.57 ± 0.23 |
| G2019S | 867 ± 110 | 0.37 ± 0.03 | 0.42 ± 0.06 |
| K1906M | 181 ± 58 | 0.28 ± 0.03 | 1.52 ± 0.52 |
| T1343A | 265 ± 25 | 0.10 ± 0.01 | 0.38 ± 0.04 |
| T1343A+ATP | 328 ± 34 | 0.10 ± 0.01 | 0.30 ± 0.03 |

All Roc autophosphorylation site mutants tested, with the exception of T1343A, showed wild-type behavior upon ATP treatment. A detailed overview of the kinetic analysis of T1343A in comparison to pathogenic variants as well as kinase-dead LRRK2 is shown in *Figure 4*. In fact, T1343A showed no increase in $K_M$ value after ATP treatment, suggesting that this is the regulatory site which is crucial for the feedback mechanism (*Figure 3C*, *Figure 3—figure supplement 1*). In addition to the lower $K_M$ value compared to wild-type that remains unaffected upon ATP treatment, also a reduced $k_{cat}$ value was observed for the T1343A mutant, which might suggest that the mutation is not completely structurally neutral. However, the T1343A mutant does not result in instable or aggregated protein (see below) and, importantly, the T1343A mutant in the RocCOR domain is the only construct that does not significantly affect the GTPase activity (*Supplementary file 1c*).

Given that LRRK2 oligomerization is an important mechanism of LRRK2 regulation, at least for LRRK2 kinase activity, we reasoned if the impact of LRRK2 autophosphorylation on the enzymatic properties of the Roc domain is mediated by changes in the oligomeric state of LRRK2. Furthermore, for the Roco protein found in the bacterium *Chlorobium tepidum*, dimerization has been demonstrated to modulate GTPase activity (*Deyaert et al., 2017*; *Gotthardt et al., 2008*). In fact, in a recent work we could demonstrate that LRRK2 dimerization is impacted by LRRK2 autophosphorylation (*Guaitoli et al., 2023*). In this work, we could demonstrate that autophosphorylation is shifting the monomer/dimer (m/d) equilibrium toward the monomeric form. Given the relevance for T1343 phosphorylation on LRRK2 GTPase activity, we wondered if the observed negative feedback might be mediated via a modulation of the m/d equilibrium. For this reason, we analyzed the relative abundance of monomers and dimers by mass photometry after ATP treatment for 30 min at 30°C (*Guaitoli et al., 2023*; *Pathak et al., 2023*). In contrast to the wild-type, for which the m/d equilibrium could be shifted toward the monomer by in vitro autophosphorylation, ATP incubation does not affect the percentage of dimer for the T1343A variant of LRRK2 (*Figure 5*). This suggests that the underlying mechanism of the negative feedback phosphorylation toward lower GTPase activity might be mediated by an increased monomerization of LRRK2.

## The phospho-null mutant T1343A is sufficient to disrupt the negative feedback in the LRRK2 signaling cascade in cells

Having identified an intramolecular negative feedback on LRRK2 GTP hydrolysis mediated by autophosphorylation, we were next interested to investigate its relevance in a cellular context. First, we analyzed the kinase activity of T1343A using an in vitro LRRKtide assay (*Jaleel et al., 2007*) to rule out that the unchanged $K_M$ and $k_{cat}$ values observed upon ATP treatment in our GTPase activity assays are the result of kinase-inactive protein. For this purpose, we compared the kinase activities for LRRK2 T1343A with the wild-type protein, as well as the hyperactive G2019S variant and a kinase-dead control. By this assay, we could clearly demonstrate that T1343A confers specific kinase activity at a level comparable to the LRRK2 wt, while G2019S was clearly more active (*Figure 6A*, *Figure 6—figure supplement 1*). Next, we performed cell-based assays, comparing the phosphorylation levels of the LRRK2 substrate Rab10. To this end, we quantified established markers (pRab10 and pS935) for LRRK2 activity by western blot analysis (*Kalogeropulou et al., 2022*; *Figure 6B*). In agreement with published data (*Kalogeropulou et al., 2022*), the other PD mutations R1441G and I2020T showed an approximately eightfold increase in Rab10



**Figure 2.** Comparison of Parkinson's disease (PD) mutants in the HPLC-based GTPase assay for the LRRK2 full-length protein. (**A**) Michaelis–Menten kinetics for the R1441G Roc-domain variant compared to LRRK2 wt. (**B**) Michaelis–Menten kinetics for the G2019S kinase-domain variant compared to LRRK2 wt. (**C**) Michaelis–Menten kinetics for a kinase-dead variant compared to LRRK2 wt.

The online version of this article includes the following figure supplement(s) for figure 2:

**Figure supplement 1.** Michaelis–Menten kinetics for MLi-2-treated LRRK2 wt.

phosphorylation (*Figure 6C and D*). Interestingly, the T1343A variant showed an approximately twofold increase in activity, which is comparable with G2019S, while the double mutant T1343A/G2019S did not further increase Rab10 phosphorylation (*Figure 6* and *Supplementary file 1*). The images used for quantification of the western blots are shown in *Figure 6—figure supplement 2*. Together, our results suggest that T1343 is crucial for the negative feedback in the LRRK2 signaling cascade in cells and the mutation to alanine disrupts this regulation. Therefore, the T1343A variant has a similar activity compared to G2019S and the double mutant T1343A/G2019S has no further increased activity.

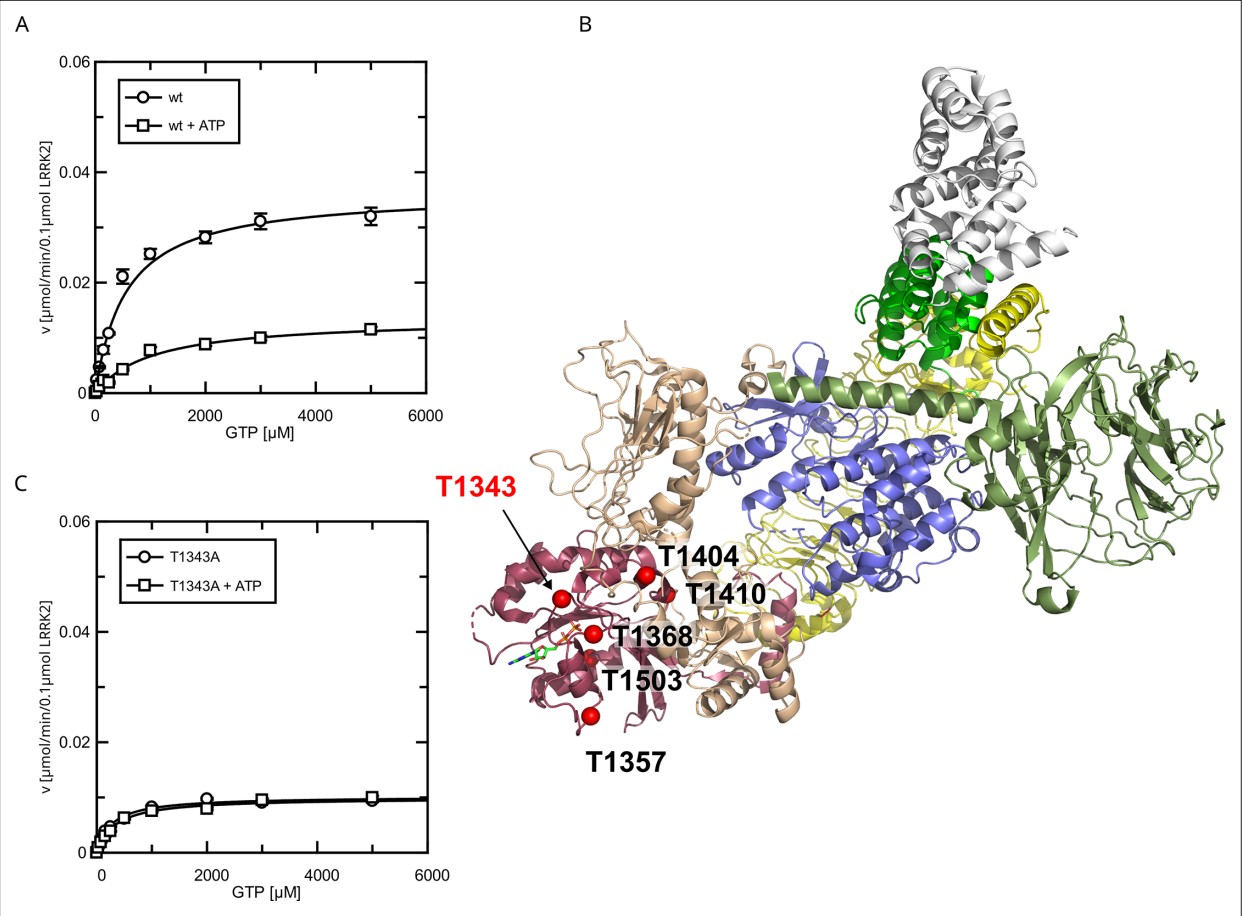

**Figure 3.** Identification of T1343 as relevant autophosphorylation site for a negative feedback loop. (**A**) Michaelis–Menten kinetics for LRRK2 wt+/-ATP, (**B**) Phosphosite screen: position of the LRRK2 phosho-sites within the Roc domain which were included in the screen mapped on PDB:7LHW (*Myasnikov et al., 2021*). Individual domains are highlighted in color as follows: Armadillo (gray), Ankyrin (green), LRR (yellow), Roc (magenta), COR (wheat), Kinase (blue), and WD40 (dark green). (**C**) Michaelis–Menten kinetics for T1343A LRRK2+/-ATP.

The online version of this article includes the following figure supplement(s) for figure 3:

**Figure supplement 1.** Initial phosphosite screen (alanine screen).

**Figure supplement 2.** AlphaFold3 models of pT1343 LRRK2, in the presence of either (**A**) GDP (Mg$^{2+}$) or (**B**) GTP (Mg$^{2+}$).

## Discussion

Although there is significant progress toward functional characterization of LRRK2 in the past two decades, including the availability of high-resolution structures covering almost the entire protein in different conformations, the precise molecular mechanisms underlying LRRK2 activation and regulation have yet to be determined (*Taymans et al., 2023*). One entry point into these mechanisms is based on conserved mechanisms in kinase-associated cellular signaling. Kinases and G-proteins are often found within the same signaling cascade, but with the exception of Roco proteins, never on the same protein. On the contrary, the usual scheme of signaling is from the G-protein via a kinase cascade toward gene expression is often regulated by the formation of signaling modules orchestrated by scaffolding proteins as given in the well-studied MAP-kinase signaling networks (*Birtwistle and Kolch, 2011*; *Kolch, 2000*). In addition, these pathways often include a negative feedback loop allowing a stable signal, a context-specific fine-tuning, as well as shutdown of the system (*Birtwistle and Kolch, 2011*). For example, an EphA2-mediated negative feedback inhibition of the Ras–PI3K–AKT module leads to a cell cycle arrest (*Menges and McCance, 2008*).

Given that LRRK2 combines a G domain with a kinase domain, resembling canonical signaling modules, besides the investigation of the kinase activity (*Taylor et al., 2020*), quite some attention has been laid on the analysis of the GTPase activity (*Biosa et al., 2013*; *Ho et al., 2016*; *Lewis*

A



B

C

**Figure 4.** Overview of the kinetic parameters for fl.LRRK2 GTPase determined by the HPLC assay. (**A**) $K_M$ values. (**B**) $k_{cat}$ values. (**C**) Catalytic efficiency ($k_{cat}/K_M$). Significant differences have been determined by an ANOVA followed by a post hoc test (n: wt=5, wt+ATP=2, R1441G=4, G2019S=4, K1906M=3, T1343A=4, T1343A+ATP=4, *p=0.05).

The online version of this article includes the following source data for figure 4:

**Source data 1.** Detailed statistical analysis.

*et al., 2007*; *Liu et al., 2010*; *Liu et al., 2016*; *Webber et al., 2011*; *Wu et al., 2019*). Nevertheless, knowledge of the interplay between these two domains is still limited. In addition, observations by most of the studies are based on single $k_{obs}$ values at considerable low GTP concentrations around or below the $K_M$ for full-length LRRK2-mediated GTP hydrolysis and/or worked with truncated LRRK2 constructs (*Liu et al., 2010*; *Wauters et al., 2018*). In this study, we have elucidated a negative feedback loop from kinase to GTPase, increasing the $K_M$ value of the Roc domain via autophosphorylation. Given an average physiological GTP concentration of approximately 500 μM within cells (*Traut, 1994*), the physiological impact of the differences in the $K_M$ values reported in this study, in combination with a negative feedback loop, is expected to lead to a strong perturbation of the tightly regulated

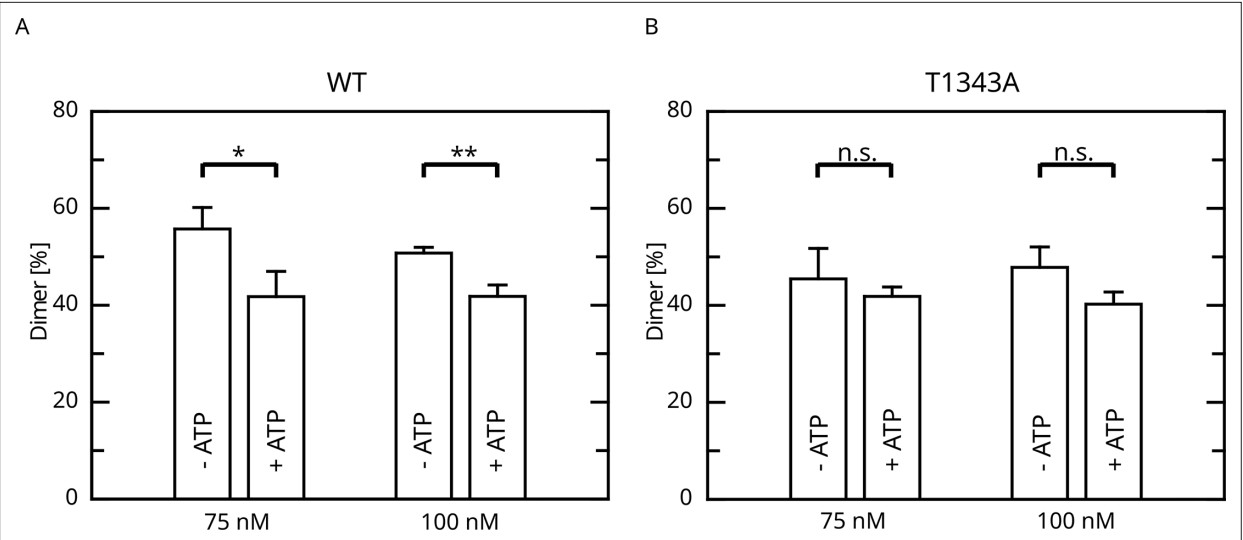

**Figure 5.** Effect of ATP incubation on the LRRK2 M/D equilibrium. (**A**) Mass photometry assays for LRRK2 wt and (**B**) T1343A LRRK2. Significance has been determined by a *t*-test (n=3, *p=0.05; **p=0.01).

LRRK2 activity. We could observe large changes in the $K_M$ induced by autophosphorylation, which supports our idea of an intramolecular negative feedback. This feedback loop has an inhibitory effect on LRRK2 Roc-mediated GTP hydrolysis. Furthermore, our study revealed that different PD variants not only alter kinase but also change the kinetics of the Roc-mediated GTP hydrolysis. In fact, when looking at the catalytic efficiency, it is striking that we found it to be increased for the R1441G variant located within a mutational hotspot of the Roc domain, also containing the PD variants R1441C, H, and S, which is in good agreement with the penetrance of these variants (*Haugarvoll et al., 2008*). In contrast, the most prevalent LRRK2 PD variant G2019S shows an increase in $K_M$, while the catalytic efficiency remains unchanged. This finding is also in well agreement with the different impact of these pathogenic variants on the Rab phosphorylation. While R1441G shows a strong Rab phosphorylation in cells, the effect mediated by G2019S is considerably weaker (*Kalogeropulou et al., 2022*). Furthermore, we could demonstrate that LRRK2 kinase activity is negatively regulating the Roc-mediated GTP hydrolysis.

By a systematic mutational analysis, removing confirmed phosphosites within the Roc domain, we identified with T1343 a critical residue, involved in the autophosphorylation-mediated negative feedback, which, in contrast to LRRK2 wt, also showed no difference in dimer–monomer formation upon ATP treatment. T1343 has initially been mapped by phosphoproteomic studies (*Gloeckner et al., 2010*; *Greggio et al., 2009*). Furthermore, it has been shown to be quantitatively phosphorylated in the C-terminal four-domain 'RCKW' construct, allowing resolving the phospho-threonine in the recently published cryo-EM structure (*Deniston et al., 2020*). T1343A and T1343G variants have already been tested in two previous studies, however, only at basal conditions and/or in vitro where similar activity was observed in comparison to wild-type LRRK2 (*Biosa et al., 2013*; *Störmer et al., 2023*). In agreement with our data, no effect of T1343G on in vitro kinase activity was observed (*Störmer et al., 2023*). Furthermore, replacing T1343 by a non-phosphorylatable residue reduced P[33] incorporation in phosphorylation assays by 50% (*Greggio et al., 2009*). As the new AlphaFold3 deep learning-based modeling software became available, recently, allowing to consider PTMs as well as small molecules, we compared the models of the GDP vs the GTP state of pT1343 LRRK2 (*Abramson et al., 2024*). Interestingly, the AF3 model suggests that the phosphate of the pT1343 is orientated inward, thereby substituting the gamma phosphate (see *Figure 3—figure supplement 2*). This finding is well in agreement with MD simulations published recently (*Störmer et al., 2023*). As we are determining GTP hydrolysis in a multi-turnover situation, the pT1343 might hamper the hydrolysis by competing with GTP re-binding. Interestingly, according to phosphoproteomes curated in the PhosphoSitePlus database (https://www.phosphosite.org) also other Rabs are phosphorylated either at a conserved serine or threonine residue at the homologous position (G1+2) within the P-loop

**Figure 6.** Effect of Roc T1343A on LRRK2 kinase activity and comparison to Parkinson's disease (PD) variants. (**A**) In vitro LRRKtide HPLC-based kinase assay (n=2). (**B**) Western blot for LRRK2 pS935, total LRRK2, Rab10 pT73, and total Rab10. (**C**) Relative Rab phosphorylation levels. (**D**) Relative LRRK2 pS935 levels. Significant differences have been determined by an ANOVA followed by a post hoc test (n=3, *p=0.05).

The online version of this article includes the following figure supplement(s) for figure 6:

**Figure supplement 1.** Raw data for the In vitro LRRKtide HPLC assay (determination of $k_{obs}$ values).

**Figure supplement 2.** Cell-based phospho-Rab assays, blot raw images (Stella imaging system, ECL+) used for quantification (ImageJ).

(for a P-loop alignment of all Rabs, see **Mishra et al., 2013**), including Rab1a, Rab5a, Rab7a, Rab8a, Rab10, Rab13, Rab15, Rab17, and Rab35, as well as Rab2 and Rab22a, respectively. In addition, also Rab4, Rab34 are phosphorylated at a P-loop residue. Furthermore, the atypical Rab protein Rab24 has a phosphorylatable tyrosine (Y17) at this position (**Ding et al., 2003**). In agreement with the findings for LRRK2 described here, Rab24 Y17A shows an increase in intrinsic GTP hydrolysis rate (**Wu et al., 2006**). Together, these findings suggest that P-loop phosphorylation is a relevant and conserved regulatory mechanism for Rab proteins.

What might be the biological consequence of this mechanism? As shown for its bacterial ortholog (*Deyaert et al., 2017*), increasing the GTP-loaded portion of LRRK2 favors its monomeric state. GTP hydrolysis leads, at least transiently, to a GDP-bound state of LRRK2, potentially inducing dimerization at a critical local protein concentration. LRRK2 kinase activity might counteract this mechanism by autophosphorylation of the P-loop residue T1343, shifting the equilibrium back to the monomeric state. This hypothesis is supported by recently determined EM structures of the bacterial LRRK2 homolog ctRoco (*Galicia et al., 2024*). This CtRoco dimer structure revealed a dimer-stabilized orientation of the P-loop, which would prevent direct nucleotide binding on the dimer. This therefore further supports a critical role of the P-loop conformation in the nucleotide-induced monomerization. Removing of this regulatory phosphosite leads to an LRRK2 activity comparable to G2019S and cannot be further increased by introducing the double mutant T1343A/G2019S. This suggests that there is a tight regulation between GTPase activity, phosphorylation, and monomer dimer equilibrium, which all interplay with each other to regulate the LRRK2 signal output. Consistently, compounds target either the kinase, G domain, or dimerization can block LRRK2 activity and signaling (*Helton et al., 2021*; *Pathak et al., 2023*; *Wojewska and Kortholt, 2021*).

However, to fully understand the LRRK2 mechanism and further explore allosteric targeting of LRRK2, a better understanding of the complex and dynamic conformational changes underlying the local orchestration and activation of LRRK2, in particular in the context of membrane localization and oligomerization as well as effector binding, such as Rab12 and Rab29 or 14-3-3 (reviewed in *Alessi and Pfeffer, 2024*), is necessary.

In conclusion, our study describes a novel intramolecular feedback mechanism of LRRK2, which is based on autophosphorylation. Similar to MAPK pathways where negative feedback mechanisms are essential to guarantee a robust signal, and also allowing a tight and context-specific shutdown of the pathway, the kinase domain in its active conformation negatively regulates GTP hydrolysis.

## Materials and methods
### DNA constructs
Cloning of the LRRK2 cDNA from human lymphoblasts is described in *Gloeckner et al., 2006*. The LRRK2 cDNA was cloned into pDONR202 vector and further subcloned by the Gateway LR-reaction (Invitrogen) into the pDEST-NSF-TAP vector for the expression of an N-terminal FLAG/tandem-STREP tag II tagged fusion protein (*Gloeckner et al., 2007*). Variants were introduced into the ENTRY construct by site-directed mutagenesis using the QuikChange II XL kit (Agilent). For the expression of the RocCOR construct in bacteria, a cDNA sequence corresponding to the LRRK2 amino acids 1293–1840 was subcloned into the pDEST-566 vector (N-terminal 6xHIS-MBP tag) using the Gateway system. pDEST-566 was a gift from Dominic Esposito (Addgene plasmid #11517; http://n2t.net/addgene:11517; RRID: Addgene_11517).

### Cell lines
HEK293T cells (CVCL_0063) were used for cell-based experiments. The genetic characteristics of this cell line were analyzed using PCR single-locus technology (Eurofins). Its identity was confirmed by comparing genetic markers with the Cellosaurus database (https://www.Cellosaurus.org). Additionally, the cell lines undergo regular testing to ensure they are free from mycoplasma contamination.

### Purification of full-length LRRK2 from HEK293T cells
The purification of NSF-tagged full-length LRRK2 has been performed as described previously with minor adaptations (*Guaitoli et al., 2016*). Briefly, HEK293T cells were transfected between 50 and 70% confluence with 8 µg of plasmid DNA/14 cm culture dish using polyethyleneimine (PEI) 25 kDa (Polysciences), and cultured post-transfection for 48 hr in 14 cm dishes in Dulbecco's Modified Eagle Medium (DMEM; Sigma-Aldrich) supplemented with 10% (vol/vol) Fetal Bovine Serum (FBS; Sigma-Aldrich) and appropriate antibiotics. After removal of the medium, the cells were resuspended in lysis buffer (1 mL/14 cm dish) containing 30 mM Tris (pH 8.0), 150 mM NaCl, 5 mM MgCl$_2$, 3 mM Dithioerythritol (DTE), 5% glycerol, and supplemented with 0.5% (v/v) Nonidet P-40 substitute, cOmplete protease inhibitor (Roche), and 0.1 mM GDP. Cell lysis was allowed to proceed for 1 hr at 4°C on a rotating shaker (10 rpm), and cell debris and nuclei were removed by centrifugation at 10,000 × *g* for

10 min. The lysate was incubated with Strep-Tactin beads (IBA, 500 µL bed volume/15 mL cell lysate) for 2 hr at 4°C on a rotating shaker. The beads were transferred to a microspin column (GE Healthcare) and washed extensively 5× with washing buffer (30 mM Tris [pH 8.0], 150 mM NaCl, 3 mM DTT, 5 mM MgCl₂, 5% [vol/vol] glycerol) containing 0.1 mM GDP. Elution was performed with 500 µL of the same washing buffer containing 2.5 mM of D-desthiobiotin (IBA) and 0.1 mM GDP. Purified proteins were further concentrated using ultracell-30 centrifugal filter units (30 kDa cutoff, Amicon) to reduce relative GDP amounts.

## Purification of 6xHIS-MBP-RocCOR from *E. coli*

6xHIS-MBP-RocCOR was purified according to standard protocols with minor adaptations (*Riggs, 1994*). Briefly, the RocCOR pDEST-566 vector was transformed in the *E. coli* BL21(DE3) strain. An overnight culture was used to inoculate into 2 L LB medium and grown at 37°C. When the culture reached an OD at 600 nm of about 0.7, protein expression was induced with 0.6 mM IPTG for 4 hr at 20°C. Cells were harvested and resuspended in resuspension buffer (50 mM HEPES pH 8, 150 mM NaCl, 10 mM MgCl₂, 10% glycerol, 0.5 mM GDP, and 5 mM β-mercaptoethanol, supplemented with 1 mM Phenylmethylsulfonyl Fluoride (PMSF) and 1× cOmplete EDTA-free protease inhibitor cocktail [Roche]), and then lysed by sonication. The cell lysate was clarified by centrifugation at 70,000 × *g*, 4°C for 1 hr. The cleared cell lysate was loaded on a 5 mL MBPTrap column (GE Healthcare). The captured proteins were washed with 5 CV wash buffer (20 mM HEPES pH 8, 200 mM NaCl, 10 mM MgCl₂, 10% glycerol, 0.5 mM GDP, and 5 mM β-mercaptoethanol), then 10 CV wash buffer supplemented with 5 mM ATP, and finally again with 5 CV wash buffer. The MBP fused protein was eluted in wash buffer containing 10 mM maltose.

## HPLC-based GTP hydrolysis assay

Steady-state kinetic measurements of LRRK2-mediated GTP hydrolysis were performed as previously described (*Ahmadian et al., 1997*). Briefly, 0.1 µM of full-length LRRK2 was incubated with different amounts of GTP (0, 25, 75, 150, 250, 500, 1000, 2000, 3000, and 5000 µM) and production of GDP was monitored by reversed-phase C18 HPLC. To this end, the samples (10 µL) were directly injected on a reversed-phase C18 column (pre-column: Hypersil Gold, 3 µm particle size, 4.6 × 10 mm; main column: Hypersil Gold, 5 µm particle size, 4.6 × 250 mm, Thermo Scientific) using an Ultimate 3000 HPLC system (Thermo Scientific, Waltham, MA) in HPLC buffer containing 50 mM KH₂PO₄/K₂HPO₄ pH 6.0, 10 mM tetrabutylammonium bromide, and 10–15% acetonitrile. Subsequently, samples were analyzed using the HPLC integrator (Chromeleon 7.2, Thermo Scientific). Initial rates of GDP production were plotted against the GTP concentration using GraFit5 (v.5.0.13, Erithacus Software). The number of experiments is indicated in the graph, and data point is the average (± s.e.m.) of indicated repetitions. The Michaelis–Menten equation was fitted to determine $K_M$ (± s.e.) and $k_{cat}$ (± s.e.).

## HPLC-based LRRKtide assay

LRRK2, LRRKtide, and ATP were mixed on ice to a final concentration 0.1 µM LRRK2, 200 µM LRRKtide, and 1 mM ATP, in 30 mM Tris/HCl pH 8.0, 150 mM NaCl, 10 mM MgCl₂, 5 mM DTE, 5% glycerol, 0.1 mM GDP. Samples were taken at 0, 5, 10, 25, 45, 65, 90, and 120 min. To stop the reaction, Trifluoroacetic Acid (TFA) was added to a final concentration of 0.1% and the samples were kept on ice before HPLC analysis. To separate phospho from non-phospho LRRKtide, the following HPLC protocol was used: Hypersil Gold C18 (2.1 × 150 mm; 3 µm particle size, Thermo Scientific), buffer A: 5% Acetonitrile (ACN), 0.1% TFA; buffer B: 80% ACN, 0.1% TFA, flow rate at 0.5 mL/min, detection at 202 nm. The elution profile was as follows: 0–1 min 100% buffer A, 1–15 min gradient up to 60% buffer B, 15–17 min 100% buffer B, 17–22 min 100% buffer A. Subsequently, samples were analyzed using the HPLC integrator (Chromeleon 7.2, Thermo Scientific). Rates of phospho-LRRKtide production were plotted against the time using GraFit5 (v.5.0.13, Erithacus Software).

## Charcoal GTP hydrolysis assay

The [γ–32P]GTP charcoal assay was performed as previously described (*Bollag and McCormick, 1995*). Briefly, 0.1 µM full-length LRRK2 or 0.5 µM 6xHIS-MBP-RocCOR was incubated with different GTP concentrations, ranging from 75 µM to 8 mM, in the presence of [γ-32P] GTP in GTPase assay buffer (30 mM Tris pH 8, 150 mM NaCl, 10 mM MgCl₂, 5% [v/v] glycerol, and 3 mM DTT). Samples were

taken at different time points and immediately quenched with 5% activated charcoal in 20 mM phosphoric acid. All non-hydrolyzed GTP and proteins were stripped by the activated charcoal and sedimented by centrifugation. The radioactivity of the isolated inorganic phosphates was then measured by scintillation counting. The initial rates of γ-phosphate release and the Michaelis–Menten kinetics were calculated as described above.

## Mass photometry (MP)

MP was performed as described in *Guaitoli et al., 2023*. Briefly, the dimer ratio of LRRK2 was determined on a Refeyn Two MP instrument (Refeyn). Prior to the experiment, a standard curve relating particle contrasts to molecular weight was established using a native molecular weight standard (Invitrogen, 1:200 dilution in HEPES-based elution buffer: 50 mM HEPES [pH 8.0], 150 mM NaCl, 5 mM MgCl$_2$, 5% [vol/vol] glycerol, 200 µM desthiobiotin, and 0.1 mM GDP). LRRK2 protein was divided into two tubes and further incubated with 0.5 mM ATP or buffer (control) for 30 min at 30°C. The LRRK2 protein was then diluted to 2× of the final concentration (end concentration: 75 nM and 100 nM) in elution buffer. The optical setup was focused in 10 µL elution buffer before adding 10 µL of the adjusted protein sample. Depending on the obtained count numbers, acquisition times were chosen between 20 s and 1 min. The dimer ratio in each measurement was normalized according to the equation

$$Normalized\ dimer\,\% = \frac{Dimer\,\%}{Monomer\,\% + Dimer\,\%}$$

Three measurements were processed for each experimental condition, and data were analyzed using GraphPad Prism.

## Cell-based Rab assay

Cell-based LRRK2 activity assays were performed as previously described (*Singh et al., 2022*). Briefly, HEK293T cells were cultured in DMEM (supplemented with 10% fetal bovine serum and 0.5% Pen/Strep). For the assay, the cells were seeded onto six-well plates and transfected at a confluency of 50–70% with SF-tagged LRRK2 variants using PEI-based lipofection. After 48 hr, cells were lysed in lysis buffer 30 mM Tris-HCl [pH 7.4], 150 mM NaCl, 1% Nonident P-40 substitute, cOmplete protease inhibitor cocktail, PhosStop phosphatase inhibitors [Roche]. Lysates were cleared by centrifugation at 10,000 × *g* and adjusted to a protein concentration of 1 µg/µL in 1× Laemmli buffer. Samples were subsequently subjected to SDS-PAGE and western blot analysis to determine LRRK2 pS935 and Rab10 T73 phosphorylation levels, as described below. Total LRRK2 and Rab10 levels were determined as a reference for normalization. For western blot analysis, protein samples were separated by SDS-PAGE using NuPAGE 10% Bis-Tris gels (Invitrogen) and transferred onto the PVDF membranes (Thermo Fisher). To allow simultaneous probing for LRRK2, on the one hand, and Rab10, on the other hand, membranes were cut horizontally at the 140 kDa MW marker band. After blocking nonspecific binding sites with 5% non-fat dry milk in TBST (1 hr, room temperature) (25 mM Tris, pH 7.4, 150 mM NaCl, 0.1% Tween-20), membranes were incubated overnight at 4°C with primary antibodies at dilutions specified below. Phospho-specific antibodies were diluted in TBST/5% Bovine Serum Albumin (BSA; Roth GmbH). Non-phospho-specific antibodies were diluted in TBST/5% non-fat dry milk powder (Bio-Rad). Phospho-Rab10 levels were determined by the site-specific rabbit monoclonal antibody anti-pRAB10(pT73) (Abcam, ab230261) and LRRK2 pS935 was determined by the site-specific rabbit monoclonal antibody UDD2 (Abcam, ab133450), both at a dilution of 1:2000. Total LRRK2 levels were determined by the in-house rat monoclonal antibody anti-pan-LRRK2 (clone 24D8; 1:10,000) (*Carrion et al., 2017*). Total Rab10 levels were determined by the rabbit monoclonal antibody anti-RAB10/ERP13424 (Abcam, ab181367) at a dilution of 1:5000. For detection, goat anti-rat IgG or anti-rabbit IgG HRP-coupled secondary antibodies (Jackson ImmunoResearch) were used at a dilution of 1:15,000 in TBST/5% non-fat dry milk powder. Antibody–antigen complexes were visualized using the ECL plus chemiluminescence detection system (GE Healthcare) using the Stella imaging system (Raytest) for detection and quantification.

## Data analysis

GTP hydrolysis was determined from metadata extracted from the chromatograms. Based on these data, Michaelis–Menten fits were performed using GraFit5 (v5.0.13). Statistical significance

of reported differences between the conditions was determined by ANOVA using Tukey post hoc test.

## Acknowledgements

The authors are grateful to Felix von Zweydorf as well as to the working students for their technical assistance in cloning the different LRRK2 variants and maintaining the plasmids. The work was supported by The Michael J Fox Foundation for Parkinson's Research (grant no: 8068.04 to CJG and AK).

## Additional information

### Funding

| Funder | Grant reference number | Author |
| --- | --- | --- |
| Michael J. Fox Foundation for Parkinson's Research | 8068.04 | Arjan Kortholt Christian Johannes Gloeckner |

The funders had no role in study design, data collection and interpretation, or the decision to submit the work for publication.

### Author contributions

Bernd K Gilsbach, Conceptualization, Formal analysis, Investigation, Methodology, Writing – original draft, Writing – review and editing; Franz Y Ho, Formal analysis, Investigation, Writing – original draft, Writing – review and editing; Benjamin Riebenbauer, Formal analysis, Investigation; Xiaojuan Zhang, Giambattista Guaitoli, Formal analysis, Investigation, Writing – review and editing; Arjan Kortholt, Conceptualization, Supervision, Funding acquisition, Writing – original draft, Writing – review and editing; Christian Johannes Gloeckner, Conceptualization, Supervision, Funding acquisition, Investigation, Writing – original draft, Writing – review and editing

### Author ORCIDs

Arjan Kortholt ⓘ https://orcid.org/0000-0001-8174-6397
Christian Johannes Gloeckner ⓘ https://orcid.org/0000-0001-6494-6944

Reviewer #1 (Public review): https://doi.org/10.7554/eLife.91083.4.sa1
Reviewer #2 (Public review): https://doi.org/10.7554/eLife.91083.4.sa2
Author response https://doi.org/10.7554/eLife.91083.4.sa3

## Additional files

### Supplementary files

• Supplementary file 1. Michaelis-Menten kinetic parameters determined for LRRK2-catalyzed GTP hydrolysis. (**a**) MBP-RocCOR Michaelis–Menten kinetics as measured by charcoal-based GTPase assay. (**b**) GTPase activity ($k_{obs}$) as measured by charcoal-based GTPase assay. (**c**) Cell-based phospho-Rab assays. Quantification of western blots.

• MDAR checklist

### Data availability

Data used for Michaelis–Menten kinetics have been deposited on Zenodo (https://doi.org/10.5281/zenodo.11242229). Raw data for western blot quantifications are shown in *Figure 6—figure supplement 2*. AlphaFold3 models have been generated under AlphaFold Server Output Terms of Use on the AlphaFold server (https://www.alphafoldserver.com, date: 5/9/2024). Final models have been deposited on Zenodo (https://doi.org/10.5281/zenodo.11242229).

The following dataset was generated:

| Author(s) | Year | Dataset title | Dataset URL | Database and Identifier |
|---|---|---|---|---|
| Gilsbach BK, FY Ho, Zhang X, Kortholt A, Gloeckner CJ | 2024 | Supplemental data for "Intramolecular feedback regulation of the LRRK2 Roc G domain by a LRRK2 kinase dependent mechanism" | https://zenodo.org/records/11242229 | Zenodo, 10.5281/zenodo.11242229 |

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
