## [Editor Report · eLife assessment]

This **valuable** article reports on the relationship between GTP hydrolysis parameters and kinase activity of LRRK2, which is associated with Parkinson's disease. The authors provide a detailed accounting of the catalytic efficiency of the ROC GTPase domain of pathogenic variants of LRRK2, in comparison with the wild-type enzyme. The authors propose that phosphorylation of T1343 inhibits kinase activity and influences monomer–dimer transitions, but the experimental evidence is currently **incomplete**.

---

## [Referee Report · Reviewer #1 (Public review)]

Summary:

This study presents careful biochemical experiments to understand the relationship between LRRK2 GTP hydrolysis parameters and LRRK2 kinase activity. The authors report that incubation of LRRK2 with ATP increases the KM for GTP and decreases the kcat. From this they suppose an autophosphorylation process is responsible for enzyme inhibition. LRRK2 T1343A showed no change, consistent with it needing to be phosphorylated to explain the changes in G-domain properties. The authors propose that phosphorylation of T1343 inhibits kinase activity and influences monomer-dimer transitions.

Strengths:

The strengths of the work are the very careful biochemical analyses and interesting results for wild type LRRK2.

Weaknesses:

The conclusions related to the involvement of a monomer-dimer transition are to this reviewer, premature and an independent method needs to be utilized to bolster this aspect of the story.

---

## [Referee Report · Reviewer #2 (Public review)]

As discussed in the original review, this manuscript is an important contribution to a mechanistic understanding of LRRK2 kinase. Kinetic parameters for the GTPase activity of the ROC domain have been determined in the absence/presence of kinase activity. A feedback mechanism from the kinase domain to GTP/GDP hydrolysis by the ROC domain is convincingly demonstrated through these kinetic analyses. However, a regulatory mechanism directly linking the T1343 phospho-site and a monomer/dimer equilibrium is not fully supported. The T1343A mutant has reduced catalytic activity and can form similar levels of dimer as WT. The revised manuscript does point out that other regulatory mechanisms can also play a role in kinase activity and GTP/GDP hydrolysis (Discussion section). The environmental context in cells cannot be captured from the kinetic assays performed in this manuscript, and the introduction contains some citations regarding these regulatory factors. This is not a criticism, the detailed kinetics here are rigorous, but it is simply a limitation of the approach.

---

## [Author Response]

The following is the authors’ response to the previous reviews.

**Public Reviews:**

**Reviewer #1 (Public Review):**
Summary:This study presents careful biochemical experiments to understand the relationship between LRRK2 GTP hydrolysis parameters and LRRK2 kinase activity. The authors report that incubation of LRRK2 with ATP increases the KM for GTP and decreases the kcat. From this they suppose an autophosphorylation process is responsible for enzyme inhibition. LRRK2 T1343A showed no change, consistent with it needing to be phosphorylated to explain the changes in G-domain properties. The authors propose that phosphorylation of T1343 inhibits kinase activity and influences monomer-dimer transitions.Strengths:Strengths of the work are the very careful biochemical analyses and interesting result for wild type LRRK2.Weaknesses:The conclusions related to involvement of a monomer-dimer transition are to this reviewer, premature and an independent method needs to be utilized to bolster this aspect of the story.

The monomer-dimer transition has been described in detail in our recent preprint Guaitoli et al., 2023 (doi: 10.1101/2023.08.11.549911). Where we in addition to mass-photometry have used blue-native page. Furthermore, to better elucidate the mechanistic impact of the phosphorylation, we have provided AlphaFold3 models. As the new AlphaFold version allows to consider PTMs as well as small molecules, we compared the models of the GDP vs the GTP-state of pT1343 LRRK2. Interestingly, the AF3 model suggests, that the phosphate of the pT1343 is orientated inwards thereby substituting the gamma phosphate (see Supplementary Figure 5). This finding is in well agreement with MD simulations published recently (Stormer et al., 2023, doi: 10.1042/BCJ20230126). As we are determining GTP hydrolysis in a multi turnover situation, the pT1343 might hamper the hydrolysis by competing with GTP re-binding. Final models have been deposited on Zenodo (https://doi.org/10.5281/zenodo.11242230).

**Reviewer #2 (Public Review):**
As discussed in the original review, this manuscript is an important contribution to a mechanistic understanding of LRRK2 kinase. Kinetic parameters for the GTPase activity of the ROC domain have been determined in the absence/presence of kinase activity. A feedback mechanism from the kinase domain to GTP/GDP hydrolysis by the ROC domain is convincingly demonstrated through these kinetic analyses. However, a regulatory mechanism directly linking the T1343 phosphosite and a monomer/dimer equilibrium is not fully supported. The T1343A mutant has reduced catalytic activity and can form similar levels of dimer as WT. The revised manuscript does point out that other regulatory mechanisms can also play a role in kinase activity and GTP/GDP hydrolysis (Discussion section). The environmental context in cells cannot be captured from the kinetic assays performed in this manuscript, and the introduction contains some citations regarding these regulatory factors. This is not a criticism, the detailed kinetics here are rigorous, but it is simply a limitation of the approach. Caveats concerning effects of membrane localization, Rab/14-3-3 proteins, WD40 domain oligomers, etc... should be given more prominence than a brief (and vague) allusion to 'allosteric targeting' near the end of the Discussion.

We thank the reviewer for the evaluation of the manuscript and suggestions made. With respect to the mentioned caveats regarding the complex regulation of LRRK2 in its native cellular environment by effectors, localization and effector binding, we have revised the discussion, accordingly. We nevertheless, want to emphasize that the phospho-null mutant T1343A leads to an increase in Rab10 phosphorylation in cells, demonstrating a relevance of this regulatory mechanism under near physiological conditions (shown in Figure 6). In addition, to further elucidate the molecular mechanisms of the p-loop phosphorylation at T1343, we have performed AlphaFold3 modelling allowing to include phosphoresidues (see comment above, Supplemental Figure 5).

Specific comments(1) The revised version is better organized with respect to the significance of monomer/dimer equilibrium and the relevance of the GTP-binding region of ROC domain that encompasses the T1343 phospho-site. The relevance of monomers/dimers of LRRK2 from previous studies is better articulated and readers are able to follow the reasoning for the various mutations.

We thank the reviewer for the positive feedback.

(2) As a suggestion I would change the following on page 6 to clarify for readers: "...would show no change in kcat and KM values upon in vitro ATP treatment" to:"...would show no change in kcat and KM values for GTP hydrolysis upon in vitroATP treatment"(3) The levels of dimer in WT (+ATP) and T1343A (+/- ATP) are the same, about 40-45%. These data are cited when the authors state that ATP-induced monomerization is 'abolished' (page 6). My suggestion is to re-phrase this conclusion for consistency with data (Fig 5). For example, one can state that 'ATP incubation does not affect the percentage of dimer for the T1343A variant of LRRK2'. This would be similar to the authors' description of these data on page 8 - 'no difference in dimer formation upon ATP treatment'.

We thank the reviewer for the suggestions. We revised the manuscript accordingly. Changes have been highlighted in the version provided for reviewing purposes.

**Recommendations for the authors:**

**Reviewer #2 (Recommendations For The Authors):**
Minor revisions-change 'Although functional work on LRRK2 has been made significant progress...' to 'Although there is significant progress toward functional characterization of LRRK2...'-change 'exact mechanisms' to 'precise mechanisms', and similarly 'exact interplay' to 'precise interplay'-change 'On a contrary' to 'On the contrary' in Discussion-change remained to be unchanged' to 'remains unchanged', page 8

We thank the reviewer for having noticed this. We have revised the manuscript accordingly.